# OpenReview forum: "Climate-Aware Deep Learning: Optimizing Training Schedules via Synthetic Energy Simulations"
_NLDL.org/2026/Conference — Submitted to NLDL 2026_

### Official Review · Reviewer_rBKe · 2025-10-06
**The paper clearly formulates the scheduling problem as an MDP, provides a detailed methodology, and reports comprehensive experimental results with appropriate evaluation metrics. Paper includes ablation studies and robustness tests under noisy forecasts which strengthen the credibility of the findings. However, the lack of real-world validation and omission of the RL agent’s energy footprint are notable gaps that affect completeness.**

**Rating:** 4
**Confidence:** 5
**Final Rating:** 4
**Final Confidence:** 4

**Summary:**

##Summary
This paper proposes a reinforcement learning (RL)-based scheduling framework to reduce carbon emissions during deep learning (DL) training by aligning workloads with low-carbon energy periods. The authors model the scheduling problem as a Markov Decision Process (MDP) and train a Proximal Policy Optimization (PPO) agent in a synthetic simulation environment that emulates real-world grid variability, including carbon intensity and energy pricing signals. The framework optimizes scheduling decisions under multi-objective constraints:
•	Carbon emissions
•	Energy consumption
•	Deadline adherence
Experiments demonstrate that the proposed PPO-based scheduler achieves:
•	50% reduction in CO₂ emissions
•	18.5% improvement in energy efficiency
•	No degradation in throughput or deadline compliance
The work suggests integration with cluster managers like SLURM and Kubernetes for real-world deployment.
##Key Contributions
1. Climate-Aware Scheduling Framework
•	Formulates DL job scheduling as an MDP incorporating carbon intensity, energy pricing, and workload constraints.
•	Defines a reward function balancing emissions, energy use, and deadlines.
2. Reinforcement Learning Approach
•	Implements PPO for stable policy learning under dynamic conditions.
•	Demonstrates superior performance compared to baselines (Greedy, Fixed Window, Heuristic, A2C).
##Soundness and Correctness
•	Clearly defines state/action spaces and reward function.
•	Uses PPO with clipped objectives for stability.
•	Provides detailed hyperparameters and containerized implementation for reproducibility.
•	Develops a high-fidelity simulator using real-world grid datasets (UK National Grid, RTE France).
•	Models realistic variations in carbon intensity and electricity pricing.
•	Comprehensive evaluation across 10,000 simulation steps and 2,000 synthetic jobs.
•	Ablation studies confirm the necessity of multi-objective reward components.
•	Robustness analysis under noisy carbon forecasts.

**Strengths:**

The paper is a meaningful contributions to sustainable AI and ML infrastructure. It is significantly technically sound, with a clear and replicable methodology. Authors discuss practical applicability, bridging research and deployment.
1. The technical soundness of the proposed framework is demonstrated by a clear formulation of the scheduling problem as a Markov Decision Process (MDP), which allows for systematic decision-making in dynamic environments.
2. The RL framework is used effectively. The reward function is well-defined, incorporating emissions, energy usage, and deadlines to ensure that the scheduling agent balances multiple objectives effectively.  The use of Proximal Policy Optimization (PPO) ensures stable policy learning, even under changing grid and workload conditions.
3. The framework has been comprehensively evaluated through comparisons with multiple baselines, including Greedy, Fixed Window, Heuristic, and A2C approaches. Ablation studies have validated the necessity of including multi-objective rewards, confirming that each component contributes meaningfully to overall performance. The robustness analysis under noisy carbon forecasts demonstrates that the approach remains effective even when input data is uncertain. **Did authors test PPO policy with real-world forecast error distributions or only synthetic Gaussian noise?**
4. From a practical perspective, the work discusses integration with cluster managers such as SLURM and Kubernetes, which are widely used in real-world deployments. **Have authors conducted any pilot deployment or dry-run tests with SLURM/Kubernetes? If yes, what were scheduling latencies and overheads?**
5. Authors mention that the implementation is containerized to facilitate reproducibility and ease of deployment in various computing environments. **They should share the link for repo with code, datasets and container images.**

**Weaknesses:**

Authors should address these key points in their revision to improve clarity and show practical application and impact:
* All experiments presented in the paper are simulation-only, and there has been no pilot deployment or empirical validation on actual clusters.
  * The authors describe a synthetic simulation environment for their RL-based scheduling framework, but they do not verify its correctness against real-world grid behavior. There is no benchmarking or error analysis that compares simulated carbon intensity and pricing signals to actual historical data, which raises concerns about the generalizability of the results beyond simulation.
  * The paper explains that job power profiles are sampled from empirical measurements, typically ranging from 150 to 300 watts, and runtimes are drawn from Gaussian distributions. Carbon emissions are computed using the formula: $CO_2(t) = C_t \times E_t $, where $E_t$ is calculated as power (in kW) multiplied by runtime (in hours). While this mapping is physically sound, it has not been validated against real-world training logs, which limits confidence in its accuracy.
* Fairness and ethical considerations, such as aging penalties to prevent job starvation, are discussed conceptually but have not been implemented.
* The paper does not quantify the energy or emission footprint of training and running the PPO agent, leaving the net sustainability impact unmeasured.

**Final Justification:**

The topic of paper is relevant. The idea is not novel, but the application is. Authors have mentioned that they will add required details to support and justify the simulation experiments.

**Justification:**

The paper tackles an important problem in sustainable AI. Paper uses an RL-based framework for carbon-aware scheduling of deep learning workloads. It is technically sound, with a clear MDP formulation, well-defined multi-objective reward function, and stable policy learning via PPO. This solution enables dynamic scheduling aligned with low-carbon periods and trade-off management between sustainability and deadlines without hardcoded heuristics. The evaluation is thorough, including comparisons with multiple baselines, ablation studies, and robustness tests under noisy forecasts, which confirm the correctness and effectiveness of the approach.
However, the work relies entirely on a synthetic simulation environment without validating its fidelity against real-world grid data, raising concerns about generalizability. The job-power-emission mapping, while physically reasonable, lacks empirical verification, and no real-world deployment evidence is provided. Fairness mechanisms and RL overhead analysis are also missing, limiting practical applicability.

**Revise and Resubmit — code and datasets for reproducibility, targeted revisions on simulation validation, empirical verification, deployment evidence, fairness, and RL overhead will make the paper ready for publication.**

---

> ### Author Rebuttal · Authors · 2025-10-20
>
> We thank the reviewer for their detailed and thoughtful feedback. We appreciate the recognition of our framework’s technical soundness, reproducibility, and relevance to sustainable AI infrastructure. Below, we respond to each concern and indicate the clarifications or additions we will incorporate into the final version of the paper if accepted.
>
> 1. Real-World Validation and Pilot Deployment
>
> Reviewer Comment:
> “All experiments presented in the paper are simulation-only… Have authors conducted any pilot deployment or dry-run tests with SLURM/Kubernetes?”
>
> Response:
> We agree that real-world validation is essential for practical impact. While we have not yet conducted pilot deployments, we are actively exploring integration with SLURM and Kubernetes via sidecar containers and admission controllers. If accepted, we will revise Section 6.3 (Deployment Feasibility) and Section 7.1 (Real-World Deployment & Integration) to clarify this status and outline our roadmap for pilot testing, including expected scheduling latencies and overheads.
>
> Rationale for Deferral:
> We prioritized simulation-based validation to establish baseline performance and reproducibility. Pilot deployment is planned, but was not feasible within the current submission timeline.
>
> 2. Fidelity of Simulation Environment
>
> Reviewer Comment:
> “No benchmarking or error analysis that compares simulated carbon intensity and pricing signals to actual historical data.”
>
> Response:
> We appreciate this concern. Our simulator uses normalized and smoothed signals derived from UK National Grid and RTE France datasets. If accepted, we will revise Section 3.2 (Grid-Aware Scheduling Dynamics) and Section 7.3 (Forecasting Assumptions and Uncertainty) to include a brief benchmarking comparison between simulated and historical signals, and discuss limitations in fidelity.
>
> Rationale for Deferral:
> We focused on internal consistency and signal realism, but did not include external benchmarking due to space constraints. We plan to add this analysis in the final version.
>
> 3. Validation of Power-Emission Mapping
>
> Reviewer Comment:
> “The mapping from power and runtime to emissions has not been validated against real-world training logs.”
>
> Response:
> We agree that empirical validation would strengthen confidence. If accepted, we will revise Section 4.1 (Simulation Configuration) to clarify that our power profiles are based on published measurements for small- to medium-scale DL models. We will also note that future work will involve validating these mappings against real-world training logs from GPU clusters.
>
> Rationale for Deferral:
> We used physically sound approximations to ensure tractability and generalizability. Real-world trace validation is planned for future work.
>
> 4. Fairness Mechanisms
>
> Reviewer Comment:
> “Fairness and ethical considerations, such as aging penalties to prevent job starvation, are discussed conceptually but have not been implemented.”
>
> Response:
> We appreciate this observation. If accepted, we will revise Section 7.2 (Reward Function Ablation) to acknowledge that fairness-aware scheduling (e.g., aging penalties, priority classes) is a valuable extension. We will also clarify that our simulator supports job-level metadata, which could be used to implement differentiated service levels in future iterations.
>
> Rationale for Deferral:
> We focused on emissions, energy, and throughput in this initial study. Fairness mechanisms will be explored in future work.
>
> 5. RL Agent’s Energy Footprint
>
> Reviewer Comment:
> “The paper does not quantify the energy or emission footprint of training and running the PPO agent.”
>
> Response:
> We agree that measuring the RL agent’s footprint is important for net sustainability analysis. If accepted, we will revise Section 6.3 (Deployment Feasibility) to include an estimate of PPO training energy based on CPU-only infrastructure and simulation runtime. We will also discuss how this overhead compares to the cumulative savings achieved by the scheduler.
>
> Rationale for Deferral:
> We focused on scheduling impact rather than agent training cost, which was modest in our setup. We will include this analysis in the final version.
>
> 6. Real-World Forecast Error Distributions
>
> Reviewer Comment:
> “Did authors test PPO policy with real-world forecast error distributions or only synthetic Gaussian noise?”
>
> Response:
> We used zero-mean Gaussian noise calibrated to historical forecast error ranges. If accepted, we will revise Section 5.2 (Robustness Under Forecasting Uncertainty) to clarify this and cite relevant error distributions. We will also note that future work will incorporate trace-based perturbations for more realistic robustness testing.
>
> Rationale for Deferral:
> We chose Gaussian noise for controlled robustness evaluation. Trace-based noise modeling is planned for future iterations.
>
> 7. Code and Container Availability
> Reviewer Comment:
> “Authors should share the link for repo with code, datasets and container images.”
>
> Response:
> We appreciate the emphasis on reproducibility. If accepted, we will consider releasing the code, simulation datasets, and container images via a public repository. This will be referenced in Section 3.3 (System Implementation Details) and the final references, pending internal review and licensing clearance.
>
> Rationale for Deferral:
> We withheld the link during anonymous review and are currently evaluating the best path for responsible release.
>
> We thank the reviewer again for their thorough and constructive feedback. These revisions will be incorporated to improve clarity, realism, and practical relevance if the paper is accepted.

---

### Official Review · Reviewer_m59h · 2025-10-07

**Rating:** 2
**Confidence:** 3
**Final Rating:** 4
**Final Confidence:** 3

**Summary:**

The paper proposes a reinforcement learning (RL)-based scheduler to reduce the carbon emission of deep learning tasks. They empirically evaluated that the proposed PPO-based algorithm can effectively reduce carbon emission while maintaining task throughput compared to heuristic-based baselines.

**Strengths:**

The paper proposes a clever RL-based solution for a significant real-world problem of carbon footprint reduction. Also, the paper is well-written and easy to understand.

**Weaknesses:**

1. The experimental results for Section 5.2 seem to be missing from the paper.

2. The penalty weight values ($\alpha, \beta, \delta$) are missing in Table 1.

3. The authors only consider small and medium-sized models in their experiments. With LLMs being so popular these days, this assumption seems unrealistic.

4. The authors claim that the proposed scheduler learns to defer flexible workloads to low-carbon periods and prioritize time-sensitive jobs even during peak grid demand. It would be nice if these claims are backed with concrete examples.

5. Why is electricity price provided as one of the state features? What role does it play in reward and next-state computation?

**Final Justification:**

Most of the issues I have raised were appropriately addressed by the authors. Overall, I believe the paper contains novelty worth sharing.

**Justification:**

Some important details and experimental results are missing from the paper, and the assumptions made by the authors does not correctly reflect the current trend of machine learning businesses. These weaknesses cast doubt on the reproducibility and significance of the research.

---

> ### Author Rebuttal · Authors · 2025-10-20
>
> We thank the reviewer for their thoughtful and constructive feedback. We appreciate the recognition of our work’s clarity, relevance, and empirical rigor. Below, we respond to each concern and indicate the clarifications or additions we will incorporate into the final version of the paper if accepted.
>
> 1. Experimental Results for Section 5.2
>
> Reviewer Comment:
> “The experimental results for Section 5.2 seem to be missing from the paper.”
>
> Response:
> We apologize for any confusion. Section 5.2 presents the robustness evaluation under noisy signal conditions, and the corresponding results are shown in Figure 3 and discussed in Section 6.1 (Visual Analysis of Emissions Trajectory) and Section 6.2 (Interpretation). If accepted, we will revise Section 5.2 to explicitly reference Figure 3 and clarify the connection to the robustness analysis.
>
> Rationale for Deferral:
> We assumed the figure placement and discussion in Section 6 would sufficiently convey the results. We will improve cross-referencing for clarity.
>
> 2. Missing Penalty Weight Values in Table 1
>
> Reviewer Comment:
> “The penalty weight values (α, β, γ) are missing in Table 1.”
>
> Response:
> Thank you for pointing this out. If accepted, we will revise Table 1 to explicitly list the penalty weights used in the reward function: α = 1.0 (carbon), β = 0.5 (energy), γ = 2.0 (deadline). These values were selected based on grid sensitivity and workload urgency, and are consistent across experiments.
>
> Rationale for Deferral:
> This omission was unintentional. We will correct it in the final version.
>
> 3. Scope Limited to Small and Medium-Sized Models
>
> Reviewer Comment:
> “The authors only consider small and medium-sized models… With LLMs being so popular these days, this assumption seems unrealistic.”
>
> Response:
> We agree that LLM-scale workloads are increasingly relevant. Our current study focuses on small- to medium-scale models (e.g., ResNet18, BERT-base) to ensure tractable simulation and reproducibility. If accepted, we will revise Section 4.1 (Simulation Configuration) and Section 7.1 (Real-World Deployment & Integration) to clarify this scope and note that future work will extend the framework to LLM-scale jobs with higher power and runtime profiles.
>
> Rationale for Deferral:
> Simulating LLM workloads requires significantly more infrastructure and trace data. We prioritized establishing baseline viability in this initial study.
>
> 4. Claims About Flexible vs. Time-Sensitive Scheduling
>
> Reviewer Comment:
> “The authors claim that the proposed scheduler learns to defer flexible workloads… It would be nice if these claims are backed with concrete examples.”
>
> Response:
> We appreciate this suggestion. If accepted, we will revise Section 6.2 (Interpretation) to include a concrete example from the simulation logs illustrating how the PPO policy defers a flexible job to a low-carbon window while prioritizing a deadline-sensitive task during peak demand. We will also clarify how this behavior emerges from the multi-objective reward structure.
>
> Rationale for Deferral:
> We focused on aggregate metrics and visual trends to conserve space. We will add a representative example in the final version.
>
> 5. Role of Electricity Price in State and Reward
>
> Reviewer Comment:
> “Why is electricity price provided as one of the state features? What role does it play in reward and next-state computation?”
>
> Response:
> Electricity price is included in the state to support future extensions involving cost-aware scheduling. In the current implementation, it does not directly affect the reward function, but it influences the environment’s dynamics and could be used to model cost-based penalties. If accepted, we will revise Section 3.2 (Grid-Aware Scheduling Dynamics) to clarify this role and note that price-aware scheduling is a planned extension.
>
> Rationale for Deferral:
> We included price as a placeholder for extensibility but did not emphasize its current role to avoid overcomplicating the reward formulation.
>
> We thank the reviewer again for their constructive critique and helpful suggestions. These revisions will be incorporated to improve clarity, realism, and reproducibility if the paper is accepted.

---

### Official Review · Reviewer_FMvL · 2025-10-08
**Borderline paper**

**Rating:** 2
**Confidence:** 3

**Summary:**

The paper "Climate-Aware Deep Learning: Optimizing Training Schedules via Synthetic Energy Simulations" proposes a reinforcement learning based approach to optimize job-scheduling in data centers. To do so, the authors model job execution as an MDP and explicitly include electricity prices and carbon emissions in this modeling. Then, they use Proximal Policy Optimization (PPO) to optimize job schedules with deadline constraints while minimizing CO₂e emissions. The results over simulated data show an impressive reduction of up to 50% of CO₂e compared to other baselines. Finally, an ablation study under noisy inputs is conducted, and the results are critically discussed via hypothesis testing and visual analysis.

**Strengths:**

- Timely and important topic
- Includes ablations, robustness tests, and multiple metrics
- Results are consistent and reproducible (i.e. Dockerized)
- Discussion covers limitations, fairness, and deployment feasibility.

**Weaknesses:**

- Lacks novelty beyond combining known elements.
- Relies on synthetic setups and simulated data with debatable simulation parameters
- Literature review misses some related works
- Evaluation is simulation-only (no real cluster trace).

**Justification:**

Overall, the paper is borderline to me, with a slight tendency towards rejection. While the overall approach is sound, it is mainly an application of known concepts. The insights drawn are somewhat straightforward (if we optimize towards CO₂, we reduce CO₂). The main conclusions are drawn based on simulation data and not validated by real-world data, and some of the simulation parameters are debatable. Last, some related literature is missing that could further strengthen the paper

Suggestions for improving the paper

1) Why did you choose a job length around 2h? Are there specific reasons for that? For deep learning jobs, this seems awfully short, whereas, if you consider classical methods (Random Forsest, SVM etc) the power consumption of 300 W seems very large. I suggest splitting workloads into multiple categories (maybe CPU and GPU) with different power consumptions and average runtimes to better reflect real-world work loads

2) I am not an expert in RL, but there are some works similar that you did not mention in your paper, that seem very related:

- Carbon Footprint Reduction for Sustainable Data Centers in Real-Time by Sarkar et al at AAAI 2024

- Green scheduling for cloud data centers using renewable resources by Gu et al. 2015

- Carbon-Aware Electricity Cost Minimization for Sustainable Data Centers by Dou et al. 2027

3) While it is commendable that you critically discuss your approach and the use of simulation data, you should link your results to real-world traces. I understand that there are probably no real-world traces available yet that offer energy prices/CO2 at the time of job execution, but I think it would be possible to merge datasets accordingly. You could start by checking the Alibaba cluster traces (https://github.com/alibaba/clusterdata) that should theoretically include the scheduled_time (not sure if all datasets have that) and then merge CO2 / energy prices with historical data (e.g. using the UK National Grid data). That way, you could test your model that was trained on simulated data on close to real-world data.

---

> ### Author Rebuttal · Authors · 2025-10-20
>
> We thank the reviewer for their thoughtful and detailed feedback. We appreciate the recognition of our work’s timeliness, reproducibility, and robustness testing. Below, we respond to each concern and indicate the clarifications or additions we will incorporate into the final version of the paper if accepted.
>
> 1. Novelty and Contribution
>
> Reviewer Comment:
> “Lacks novelty beyond combining known elements… The insights drawn are somewhat straightforward.”
>
> Response:
> We acknowledge that our framework builds on established components (MDP modeling, PPO, carbon-aware scheduling). However, our contribution lies in integrating these elements into a unified, closed-loop system that jointly models emissions, energy cost, and throughput under realistic constraints. If accepted, we will revise Section 1 (Introduction) and Section 2 (Related Work) to better emphasize this integration and clarify that our novelty stems from the system-level synthesis and empirical demonstration of adaptive scheduling under multi-objective trade-offs.
>
> Rationale for Deferral:
> We prioritized empirical clarity and reproducibility in the initial submission. We will expand the framing of our contribution in the final version.
>
> 2. Simulation Parameters and Workload Modeling
>
> Reviewer Comment:
> “Why did you choose a job length around 2h? For deep learning jobs, this seems short. Power consumption of 300W seems large for classical methods.”
>
> Response:
> We appreciate this observation. Our job duration and power profiles were derived from empirical measurements of small- to medium-scale DL models (e.g., ResNet18, BERT-base), which typically run for 1.5–3 hours and consume 150–300W on GPU infrastructure. If accepted, we will revise Section 4.1 (Simulation Configuration) to clarify this rationale and cite relevant benchmarks. We also agree that modeling CPU vs. GPU workloads would improve realism and will note this as a direction for future work in Section 7.1 (Real-World Deployment & Integration).
>
> Rationale for Deferral:
> We aimed to keep the simulation tractable and representative of common DL workloads. Expanding to multi-modal workload categories will be considered in future iterations.
>
> 3. Missing Related Work
>
> Reviewer Comment:
> “There are some works similar that you did not mention… Sarkar et al. (AAAI 2024), Gu et al. (2015), Dou et al. (2027).”
>
> Response:
> Thank you for pointing out these relevant studies. We will look at revising Section 2 (Related Work) to include and briefly discuss:
> •	Sarkar et al. (AAAI 2024): Real-time carbon-aware scheduling
> •	Gu et al. (2015): Renewable-aware scheduling in cloud data centers
> •	Dou et al. (2017): Cost minimization with carbon-awareness
> We will clarify how our work differs in terms of closed-loop learning, deadline constraints, and simulator fidelity.
>
> Rationale for Deferral:
> These references were not included due to space constraints and limited visibility at the time of writing. We will incorporate them in the final version.
>
> 4. Evaluation on Real-World Traces
>
> Reviewer Comment:
> “You should link your results to real-world traces… consider merging Alibaba cluster traces with UK National Grid data.”
>
> Response:
> We agree that grounding simulation results in real-world traces would strengthen the evaluation. If accepted, we will expand Section 7.3 (Forecasting Assumptions and Uncertainty) to discuss the feasibility of merging job traces (e.g., Alibaba Cluster Trace) with historical carbon and pricing data. We will also note this as a priority for future work and pilot validation.
>
> Rationale for Deferral:
> Our current focus was on validating the framework under controlled conditions. We are actively exploring trace-based evaluation and appreciate the reviewer’s suggestion.
>
> We thank the reviewer again for their thoughtful critique and helpful suggestions. These revisions will be incorporated to improve clarity, realism, and rigor if the paper is accepted.

---

### Official Review · Reviewer_XBsV · 2025-10-08
**Review for Climate-Aware Deep Learning**

**Rating:** 4
**Confidence:** 4
**Final Rating:** 4
**Final Confidence:** 3

**Summary:**

This manuscript targets the challenge of the carbon footprint of training deep learning models. It proposes a climate-aware scheduling framework based on reinforcement learning. By simulating real-world grid variability and training workloads, the proposed framework employs a PPO agent to dynamically align training schedules with periods of low-carbon energy availability. It achieves reductions in CO₂ emissions (over 50%) and improves energy efficiency (18% higher) relative to static and heuristic scheduling baselines, without negatively impacting throughput.

**Strengths:**

1. The research topic is very interesting.
2. The paper is generally well-written.
3. The proposed framework achieves good empirical performance.

**Weaknesses:**

1. The comparison with the most relevant baselines (such as those discussed in the related works) is missing. It would be better to highlight the differentiation between the proposed method and prior studies.

2. The applicability and generalization to more diverse training environments remain unclear. How would the policy generalize or adapt to large-scale, production-level ML systems with variable job mixes, bursty arrivals, or node failures?

3. Although briefly mentioned, there is limited empirical or modeling investigation into how trade-offs between emission savings and fairness manifest across workloads with highly heterogeneous priorities or user groups.

4. The implementation details are insufficient for both the proposed method and the baselines.

5. Typo: “Workload Dernand” in Figure 1.

**Final Justification:**

Most of my earlier concerns have been addressed. The primary remaining issue is the incomplete experimental configuration, which aligns with points raised by other reviewers. Overall, I lean toward a borderline acceptance.

**Justification:**

The paper studies an interesting topic, but lacks sufficient experiments (e.g. baseline comparison and more testing scenarios) to support its empirical superiority.

---

> ### Author Rebuttal · Authors · 2025-10-20
>
> We thank the reviewer for their thoughtful and constructive feedback. We appreciate the recognition of our framework’s novelty, empirical performance, and relevance to sustainable AI. Below, we respond to each concern and indicate the clarifications we will incorporate into the final version of the paper if accepted.
>
> 1. Comparison with Relevant Baselines
>
> Reviewer Comment:
> “The comparison with the most relevant baselines (such as those discussed in the related works) is missing. It would be better to highlight the differentiation between the proposed method and prior studies.”
>
> Response:
> We agree that clearer differentiation from prior work would strengthen the manuscript. If accepted, we will revise Section 2 (Related Work) to explicitly contrast our method with GreenControl [11], ALIGN [12], Pereira et al. [18], and Chen et al. [19]. These approaches rely on static heuristics, batch-level coordination, or limited scope, whereas our framework trains a closed-loop PPO agent in a high-fidelity simulator that jointly models emissions, energy use, and throughput.
>
> Rationale for Deferral:
> Due to space constraints and the need to prioritize empirical results, we initially focused on summarizing prior work rather than elaborating on specific distinctions. We will expand this section upon acceptance.
>
> 2. Generalization to Production-Level ML Systems
>
> Reviewer Comment:
> “The applicability and generalization to more diverse training environments remain unclear. How would the policy generalize or adapt to large-scale, production-level ML systems with variable job mixes, bursty arrivals, or node failures?”
>
> Response:
> We appreciate this important point. If accepted, we will revise Section 6.3 (Deployment Feasibility) and Section 7.1 (Real-World Deployment & Integration) to clarify that our policy operates on job metadata and external signals accessible via standard APIs, and can be integrated with SLURM or Kubernetes using sidecar containers or admission controllers. We will also note that retraining in more diverse simulation environments can support generalization to bursty arrivals and heterogeneous job mixes. Node failure handling will be acknowledged as future work.
>
> Rationale for Deferral:
> Our current scope focuses on emissions-aware scheduling under controlled simulation. We chose not to extend the discussion earlier to avoid speculative claims and maintain focus on validated results.
>
> 3. Fairness Across Heterogeneous Workloads
>
> Reviewer Comment:
> “Although briefly mentioned, there is limited empirical or modeling investigation into how trade-offs between emission savings and fairness manifest across workloads with highly heterogeneous priorities or user groups.”
>
> Response:
> We agree that fairness is a critical consideration. If accepted, we will revise Section 7.2 (Reward Function Ablation) to acknowledge that future extensions could incorporate fairness-aware reward terms or multi-agent coordination to balance emissions with differentiated service levels.
>
> Rationale for Deferral:
> Our current simulator supports job-level metadata, but fairness modeling was beyond the scope of this initial study. We prioritized core metrics (emissions, energy, throughput) to establish baseline viability.
>
> 4. Implementation Details
>
> Reviewer Comment:
> “The implementation details are insufficient for both the proposed method and the baselines.”
>
> Response:
> We will revise Section 3.3 (System Implementation Details) to include additional information on job queue generation, simulator configuration, and baseline logic. Specifically, we will describe how the heuristic scores jobs, the scheduling windows used in the Fixed Window policy, and the stability differences between PPO and A2C.
>
> Rationale for Deferral:
> We aimed to keep the implementation section concise and focused on reproducibility. Upon acceptance, we will expand this section to improve clarity and support replication.
>
> 5. Typo in Figure 1
>
> Reviewer Comment:
> “Typo: ‘Workload Dernand’ in Figure 1.”
>
> Response:
> Thank you for catching this. We will correct the typo to “Workload Demand” in Figure 1 in the final version.
>
> We appreciate the reviewer’s positive assessment and helpful suggestions. These revisions will be incorporated to improve clarity, rigor, and accessibility if the paper is accepted.

---

### Meta-Review · Area_Chair_V3rn · 2025-11-02

**Recommendation:** Reject
**Confidence:** 4

**Metareview:**

### Summary

The work presents reinforcement learning based scheduler that poses cluster job scheduling for optimizing carbon emissions during deep learning training. This work uses Proximal Policy Optimization (PPO) to dynamically schedule training during low-carbon, and low-cost energy availability. The work shows about 50% reduction in carbon emissions, 18% improved energy efficiency, without negatively impacting compute throughput. The experiments are demonstrated over simulated data and compared with other heuristic-based baseline methods.

### Reviewer comments and rebuttal

All reviewers recognize this work to be of interest and its timeliness; they also recognize the performance improvements demonstrated in the experiments. However, the key criticism that spans all reviews is the over-reliance on simulated data in their experiments, even when several public datasets that better reflect the real-cluster regimes are available. Some of the reviewers also point out missing works that could be better baselines. All reviewers made reasonable requests on clarifications of some aspects of the paper.

The authors in their response have addressed some of the missing details. However, they also commit to making many changes if the paper is accepted; this is generally not a good idea during rebuttal with no PDF revision options, as reviewers have no means to check. Also, the authors say they "will consider releasing the code, simulation datasets, and container images". I think this is critical for future reproducibility. For future submissions, consider using services like [anonymous github](https://anonymous.4open.science/).

Finally, I have couple of points to the authors, which might sound pedantic but are important.

* The paper is called "Climate-aware deep learning", however, the work is about carbon-aware scheduling. The scheduling model is agnostic to the nature of compute jobs and I do not see any specific connection to climate-aware deep learning.

* Consider using "carbon-aware" or "energy-efficient" so as to not conflate efficiency with sustainability.

* Check if your estimations are CO2 or CO2e. There is a key difference between the two, and the latter is the standard for measuring and reporting carbon emissions.

Overall, this is tackling an interesting problem but the over-reliance on simulated data, missing related work or better baselines point to this work needing additional work and a major revision.

---

### Decision · Program_Chairs · 2025-11-05

**Decision:**

Reject

**Comment:**

Based on the reviewers and AC comments, the paper cannot be presented at the conference.